# Clinicopathological features and ethnic disparities of melanoma in the United Arab Emirates: 2017–2025

Jonathan Mokhtar[1]*, Nada Alsuwaidi[1], Nada Hassane[1],
Rose Mary Eapen[2], Hind Aljanaahi[1], Dalia AlDhamin[1], Tannaz Rahbari[1],
Ghazal Talal Saeed[1], Zainab Abdulla Al Darwish[3], Sara Almalik[1],
Jeyaseelan Lakshmanan[1], Tom Loney[1], Reem El-Bahtimi[2]

1 Mohammed Bin Rashid University of Medicine and Health Sciences, College of Medicine (MBRU-CoM), Dubai, United Arab Emirates, 2 International Dermpath Consult FZ-LLC, Dubai, United Arab Emirates, 3 Department of Dermatology and Aesthetic Center, Rashid Hospital, Dubai Health, Dubai, United Arab Emirates

☉ These authors have contributed equally to the manuscript.
* Jonathan.mokhtar@students.mbru.ac.ae

## Abstract

### Background

Cutaneous melanoma incidence is rising globally, yet clinicopathological data from the high ultraviolet (UV) environment in the United Arab Emirates (UAE), with its diverse expatriate population, remain scarce. This study aims to characterize the histopathological features of cutaneous melanoma in a large, multi-ethnic cohort in the UAE.

### Methods

This retrospective cross-sectional study analyzed histopathologically confirmed cases of cutaneous melanoma diagnosed at a specialized dermatopathology laboratory in the UAE from 01/01/2017–01/01/2025. Patient demographics, tumor location, histologic subtype, Clark level, Breslow thickness, mitosis, lymphovascular, and perineural invasion were extracted and analyzed. Descriptive statistics, group comparisons, and multivariable logistic regression were performed using IBM SPSS version 29.0 to identify predictors of thick melanoma (Breslow thickness >1.0 mm).

### Results

A total of 597 patients met the inclusion criteria (50.8% male; mean age 47.4 ± 12.3 years). Individuals of European ancestry constituted 73.4% of cases. Superficial spreading melanoma was the predominant subtype (58.5%), and 46.9% of melanomas were thin (≤1.0 mm). Males presented with significantly thicker tumors than females (Breslow thickness of 0.72 ± 1.32 vs. 0.50 ± 0.58 mm; $p < 0.01$) and exhibited

**Data availability statement:** All relevant data are within the paper.

**Funding:** No external financial support nor grants were received from any public, commercial, or not-for-profit entities for the research, authorship, or publication of this study. The authors would like to thank Mohammed Bin Rashid University of Medicine and Health Sciences for covering the article processing charge for this paper.

**Competing interests:** I have read the journal's policy and the authors of this manuscript declare no competing interest of any type.

distinct anatomical distributions predominant to the trunk as compared to females with leg predominance. Multivariable analysis confirmed nodular melanoma (OR 18.40; 95% CI [7.08, 47.86]; $p < 0.001$) as the single strongest independent predictor of thick melanoma.

## Conclusion

Melanoma in the UAE disproportionately affects fair-skinned expatriates and frequently presents with sex-specific clinical patterns. These findings highlight the need for targeted public awareness initiatives to reduce melanoma morbidity and mortality in the region.

## 1. Introduction

Melanoma, a malignant neoplasm of melanocytes, represents a significant and escalating global health challenge. Despite many cases being preventable, it remains the most lethal form of skin cancer worldwide [1]. The global burden of melanoma is substantial, with an estimated 325,000 new cases and 57,000 deaths reported in 2020 alone, with recent GLOBOCAN estimates indicating that in 2025, approximately 107,960 cases of cutaneous melanoma were diagnosed, with 8,430 deaths occurring worldwide [2]. Projections indicate a concerning trajectory, with the number of new cases expected to rise by approximately 50% to 510,000 and deaths to increase by 68% to 96,000 by 2040 if current trends persist [2]. The incidence of melanoma exhibits substantial geographic variation, with the highest rates observed in sun-exposed populations in Australia and New Zealand, followed by North America and Europe [2,3]. Conversely, melanoma has historically been considered rare in most African and Asian countries, including the Middle East, with incidence rates often below 1 per 100,000 person-years [2].

However, this traditional view of melanoma as a low-incidence cancer in the Middle East is being challenged by emerging data [4,5]. While research output from the region remains limited, studies suggest a growing number of cases, highlighting a potential shift in the region's epidemiological landscape [4]. In the United Arab Emirates (UAE), a nation characterized by its multicultural expatriate population, year-round sun exposure, and rapid urbanization, skin cancer now ranks as the fourth most common malignancy [5]. This rising incidence occurs within a context of significant barriers to early detection and prevention.

The confluence of a rising cancer burden and low public awareness creates a critical public health issue. The unique demographic profile of the UAE, with its diverse mix of skin phenotypes and cultural attitudes towards sun exposure, presents a complex scenario for which there is a paucity of specific data. Understanding the distinct clinicopathological patterns and risk factors of melanoma in this population is essential for developing effective, targeted public health interventions, awareness campaigns, and clinical screening strategies. Therefore, this study was conducted to investigate melanoma patterns in the UAE, aiming to raise awareness among healthcare practitioners, especially dermatologists, to screen patients for melanoma.

## 2. Materials and methods

### 2.1 Study design and setting

This study was conducted as a retrospective cross-sectional analysis, in accordance with the Strengthening the Reporting of Observational Studies in Epidemiology (STROBE) guidelines (S1 File) [6]. All patients with histopathologically confirmed cutaneous melanoma diagnosed during the eight years from 01/01/2017–01/01/2025 were retrospectively identified. Data were accessed for research purposes on 10/02/2025. Data access was restricted to the senior authors, and all records were fully de-identified before IRB approval and analysis.

The study was performed at a single, high-volume dermatopathology laboratory in the UAE (International DermPath Consult FZ-LLC), a setting characterized by a highly diverse, multinational patient population with over 60 facilities, including private hospitals, polyclinics, and patients for second opinion. The study adhered to the ethical principles of the Declaration of Helsinki. Ethical approval was granted by the Dubai Scientific Research Ethics Committee (DSREC) (Reference number: DSREC-11/2024_41), which waived the requirement for informed consent due to the study's retrospective nature and the use of anonymized pathology reports.

### 2.2 Study population and participant selection

The population included all patients who received a new, primary diagnosis of cutaneous melanoma, confirmed by histopathological examination, at our institution during the defined study period. Patients with both invasive melanoma and melanoma in situ were eligible for inclusion to ensure a complete profile. Case identification was performed by the senior author (R.E.-B.), who systematically searched the institution's electronic pathology database using specified diagnostic codes corresponding to all subtypes of cutaneous melanoma. To ensure the integrity of the cohort, we applied specific exclusion criteria. Patients were excluded if they were diagnosed with non-melanoma skin cancer, presented with metastatic melanoma from an unknown primary site, or had medical records that were incomplete or inaccessible, thereby precluding the reliable extraction of key demographic and clinicopathological variables.

### 2.3 Diagnostic procedures and histopathological assessment

Initial clinical assessment of suspicious pigmented lesions in the UAE was performed by licensed dermatologists using the ABCDE criteria (Asymmetry, Border irregularity, Color variegation, Diameter >6mm, Evolving) [7]. Patients with suspicious moles underwent excisional biopsy and fixation in 10% neutral buffered formalin. Upon receipt in the pathology laboratory, specimens were accessioned, serially sectioned, and entirely submitted for processing. The tissue was then embedded in paraffin, cut in 4–5 micrometer-thick (μm; $10^{-6}$ m) sections and stained with hematoxylin and eosin. For diagnostically challenging cases, immunohistochemical staining was performed using a panel of melanocytic markers, including S0X10, Melan-A, p16, and HMB-45, to confirm the diagnosis and differentiate melanoma from other pigmented lesions. PRAME immunostain was not available at the time. All the cases were examined by a certified member of the American Board of Dermatopathology (R.E.-B.).

### 2.4 Data collection and variable definitions

The senior author (R.E.-B.) developed a standardized, de-identified data collection instrument in Microsoft Excel to safeguard patient privacy and data integrity. All relevant variables were systematically extracted from the Laboratory information system (LIS) and corresponding definitive pathology reports with synoptic reports. Data extraction was performed independently by two authors (J.M. and R.E.-M.) using the de-identified Microsoft Excel sheet to ensure accuracy and minimize transcription errors, with any discrepancies adjudicated by the senior author (R.E.-B.). Demographic characters

were recorded as follows: age at diagnoses, captured as a continuous variable (in years), sex was documented as male or female, and self-reported ethnicity was classified into six geographically based categories according to World Health Organization (WHO) designations: European, Australasian, Middle Eastern and North African (MENA), African, South and Southeast Asian, and South American.

All tumor characteristics were based on the final official histopathological report. The histological subtype was classified according to the 2018 WHO classification of skin tumors, including superficial spreading melanoma (SSM), nodular melanoma (NM), lentigo maligna melanoma (LMM), melanoma in situ, and a category for other rare or unclassifiable subtypes (including acral lentiginous melanoma [ALM], desmoplastic melanoma, mucosal melanoma, and spitzoid melanoma) [8]. The primary prognostic indicator, Breslow thickness, was recorded in millimeters (mm) and represents the maximal vertical tumor thickness measured from the top of the granular layer of the epidermis to the deepest point of tumor invasion. Breslow thickness was categorized into four clinically relevant groups based on the American Joint Committee on Cancer (AJCC) staging system: ≤ 1.0 mm (thin), 1.01–2.0 mm, 2.01–4.0 mm, and >4.0 mm (very thick) [9]. The Clark level of invasion, representing the anatomical plane of invasion through the layers of the skin, was recorded as levels I–V [9]. Finally, the tumor's primary anatomical location was documented and grouped into four major body sites: head and neck, back and torso, arm and hand, and leg and foot.

Additional histopathological features associated with melanoma prognosis were also extracted from pathology reports, including ulceration, mitotic count/mm$^2$, lymphovascular invasion (LVI), and perineural invasion (PNI). Ulceration was defined as the absence of an intact epidermis overlying the tumor. Mitotic count was recorded based on the presence of mitotic figures identified in the dermal component of the tumor/mm$^2$. LVI and PNI were recorded when tumor cells were identified within vascular or lymphatic channels or within or surrounding nerve fibers, respectively.

## 2.5 Bias

Potential biases were addressed through several strategies. Selection bias was minimized by including data from diverse healthcare facilities across several Emirates within the UAE, representing a wide range of demographic groups. To ensure comparability across patient groups, all histopathological analyses were conducted centrally at the International DermPath Consult, FZ-LLC, Dubai, UAE, using standardized protocols, thereby reducing inter-observer variability. Observer bias was further reduced by involving an independent epidemiologist and biostatistician (T.L. and J.L.), who performed the statistical analysis using the de-identified dataset linked only by unique identification numbers, without access to any direct patient identifiers.

## 2.6 Statistical analysis

All statistical analyses were conducted using SPSS Statistics for Windows, Version 29.0 (IBM Corp., Armonk, NY). The cohort's characteristics were summarized using descriptive statistics. Continuous variables that were normally distributed were reported as mean ± standard deviation (SD). Categorical variables, including sex, ethnicity, histological subtype, and anatomical location, were reported as frequencies and percentages (%).

To evaluate differences in clinical presentation between sexes, group comparisons were performed. An independent-samples t-test was used to compare the means of normally distributed continuous variables (age and Breslow depth) between males and females. For categorical variables, the chi-square ($\chi^2$) test was employed; in cases where the expected cell count was less than five, Fisher's exact test was used to ensure statistical validity. A two-sided p-value <0.05 was considered statistically significant for all analyses.

To identify independent factors associated with more advanced disease at presentation, a multivariable logistic regression analysis was conducted. The primary outcome for this model was the presence of a thick melanoma, defined as Breslow depth >1.0 mm. The dependent variable was therefore dichotomized into thin (≤1.0 mm) and thick (>1.0 mm) tumors. Covariates selected for inclusion in the model were based on clinical relevance and included age group, sex,

primary anatomical body site, and histological subtype. The results of the regression analysis were presented as odds ratios (ORs) with their corresponding 95% confidence intervals (CIs). The goodness-of-fit of the final logistic regression model was formally assessed using the Hosmer-Lemeshow test.

## 3. Results

### 3.1 Participants

Over the eight-year study period, 597 patients met the inclusion criteria and were included in the final analysis (Fig 1). Reasons for exclusion of other participants included the following: metastatic disease, dysplastic nevi, re-excisions with no residual melanoma and other diagnoses not related to melanoma. The cohort was almost evenly split by sex, comprising 303 males (50.8%) and 294 females (49.2%). The baseline demographic and clinical characteristics of the study population are summarized in **Table 1**.

### 3.2 Demographic profile

The average age at diagnosis for the entire cohort was 47.4 ± 12.3 years. A statistically significant difference in age at diagnosis was observed between the sexes, with males presenting at a mean age of 50.6 ± 11.9 years, approximately 6.5

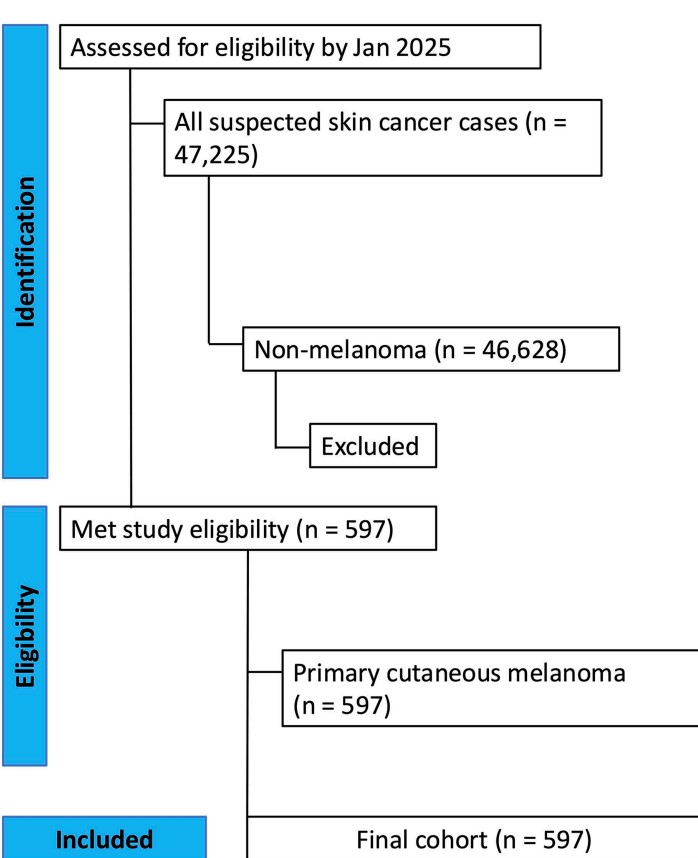

**Fig 1. STROBE Flow diagram demonstrating identification, screening, and inclusion of patients with primary cutaneous melanoma.**

**Table 1. Baseline characteristics of the study population.**

| | Total (n = 597) | Males (n = 303) | Females (n = 294) |
|---|---|---|---|
| Age (years) | 47.4 ± 12.3[a] | 50.63 ± 11.90[a] | 44.05 ± 11.97[a] |
| Ethnicity (n) | | | |
| European | 406 | 212 | 194 |
| MENA | 65 | 32 | 33 |
| African | 33 | 12 | 21 |
| Australasian | 28 | 21 | 7 |
| South/Southeast Asian | 11 | 3 | 8 |
| South American | 10 | 5 | 5 |
| Other | 44 | 18 | 26 |
| **Clark Level (n)** | | | |
| I | 121 | 58 | 63 |
| II | 221 | 104 | 117 |
| III | 172 | 93 | 79 |
| IV | 82 | 47 | 35 |
| V | 1 | 1 | 0 |

**Abbreviations:** MENA: Middle East and North Africa, [a]mean ± standard deviation

years older than females, who had an average age of 44.1 ± 12.0 years ($p < 0.001$), possibly suggesting delayed presentation due to lower dermatological awareness and/or different sun exposure safety habits amongst them.

The cohort was ethnically diverse, though predominantly composed of individuals of European descent (n = 406, 73.4%). The next largest group was patients from the MENA region (n = 65, 11.8%), followed by smaller representations from African (n = 33, 6.0%), Australasian (n = 28, 5.1%), South and Southeast Asian (n = 11, 2.0%), South American (n = 10, 1.8%), and other (n = 44, 7.4%) populations.

### 3.3 Outcome data

**Histopathologic features.** The most prevalent histological subtype was SSM, accounting for 58.5% of all cases. This was followed by melanoma in situ (14.7%), LMM (13.2%), and NM (9.7%). While SSM was the most common subtype in both sexes, NM was notably more frequent in males (12.2%) compared to females (7.1%) (Fig 2).

Ulceration, mitotic activity, LVI, and PNI were also evaluated across melanoma subtypes. NM demonstrated the highest frequency of adverse histopathologic features, particularly ulceration and mitotic activity, whereas PNI was most observed in desmoplastic melanoma. LVI was rare and identified in only a single case. The distribution of these histopathologic features, overall and stratified by sex, is summarized in **Table 2**.

**Tumor invasion and thickness.** The distribution of the Clark invasion level is shown in **Table 1**. Most tumors were classified as Clark level II (37.0%) or III (28.9%). Regarding Breslow thickness, nearly half of the tumors (46.9%) were classified as thin (≤1.0 mm), with an average thickness of 0.62 ± 1.03 mm. However, males had, on average, significantly thicker tumors than females (Breslow depth: 0.72 ± 1.32 mm vs. 0.50 ± 0.58 mm; $p < 0.01$). This trend is further evident in the categorical distribution, which shows a higher proportion of males with tumors >1.0 mm in thickness (Fig 3).

**Anatomical location.** The anatomical distribution of melanoma showed distinct patterns between sexes, as detailed in **Fig 4**. The most common site of melanoma in females was the leg and foot (37.8%), whereas in males it was the back and torso (43.9%). Males also had a higher proportion of melanomas on the head and neck (15.8% vs. 8.8%).

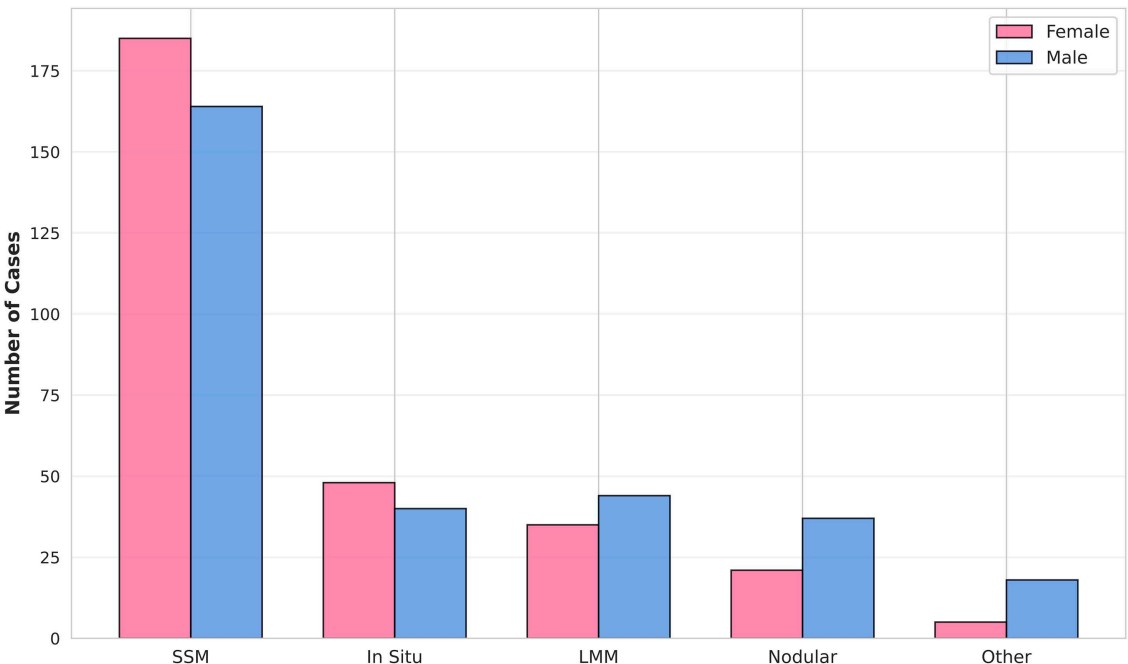

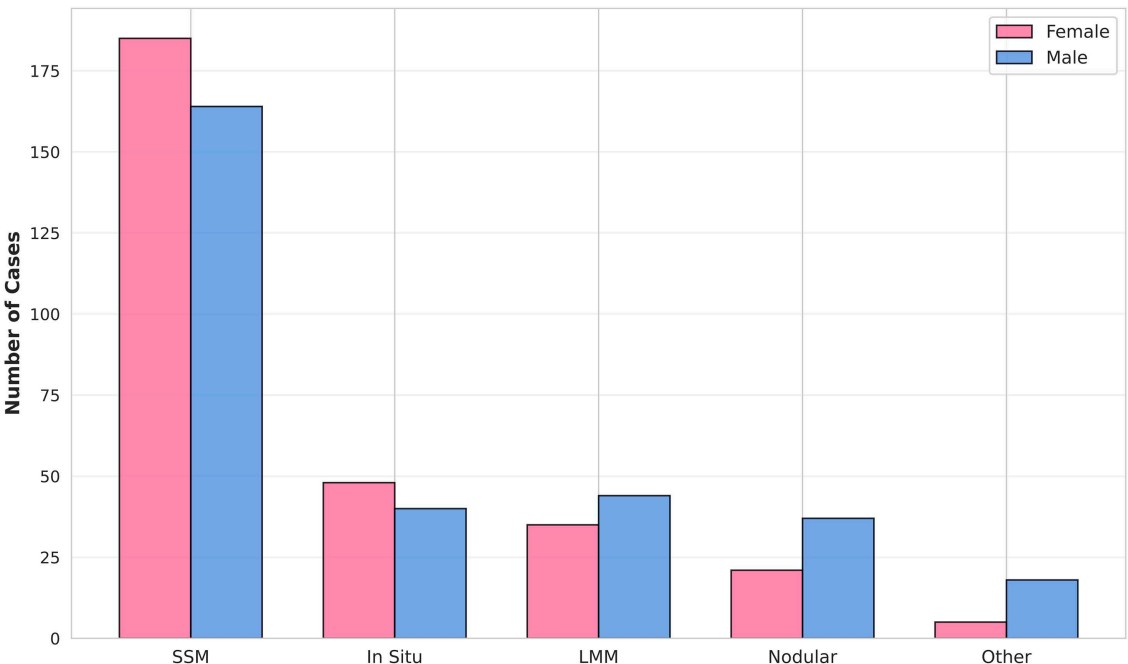

**Fig 2. Distribution of Melanoma Histologic Subtypes by Sex.** Other subtypes include acral lentiginous melanoma, desmoplastic melanoma, and spitzoid melanoma.

**Table 2. Distribution of the histopathologic prognostic features across melanoma subtypes.** The table summarizes the frequency of ulceration, mitotic activity, lymphovascular invasion (LVI), and perineural invasion (PNI) across major histologic subtypes, including nodular melanoma (NM), superficial spreading melanoma (SSM), acral lentiginous melanoma (ALM), and desmoplastic melanoma.

|  | NM | SSM | Acral | Mucosal | Desmoplastic |
|---|---|---|---|---|---|
| **Total** |  |  |  |  |  |
| Ulceration | 17 | 1 | 1 | 1 | 0 |
| Mitosis | 41 | 2 | 4 | 2 | 1 |
| LVI | 1 | 0 | 0 | 0 | 0 |
| PNI | 0 | 1 | 0 | 0 | 2 |
| **Males** |  |  |  |  |  |
| Ulceration | 12 | 0 | 0 | 1 | 0 |
| Mitosis | 31 | 0 | 2 | 1 | 1 |
| LVI | 0 | 0 | 0 | 0 | 0 |
| PNI | 0 | 1 | 0 | 0 | 1 |
| **Females** |  |  |  |  |  |
| Ulceration | 5 | 1 | 1 | 0 | 0 |
| Mitosis | 10 | 2 | 2 | 1 | 0 |
| LVI | 1 | 0 | 0 | 0 | 0 |
| PNI | 0 | 0 | 0 | 0 | 1 |

Abbreviations: NM: Nodular Melanoma, SSM: Superficial Spreading Melanoma, ALM: Acral Lentiginous Melanoma, LVI: Lymphovascular Invasion, PNI: Perineural Invasion.

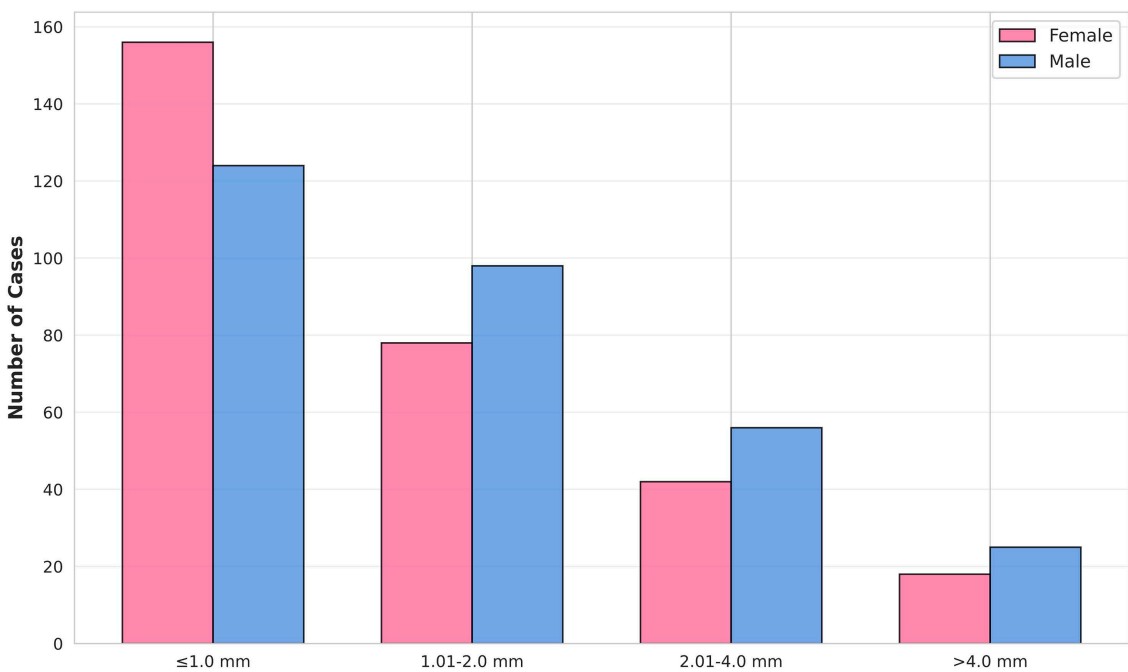

**Fig 3. Breslow thickness categories of melanoma by sex.**

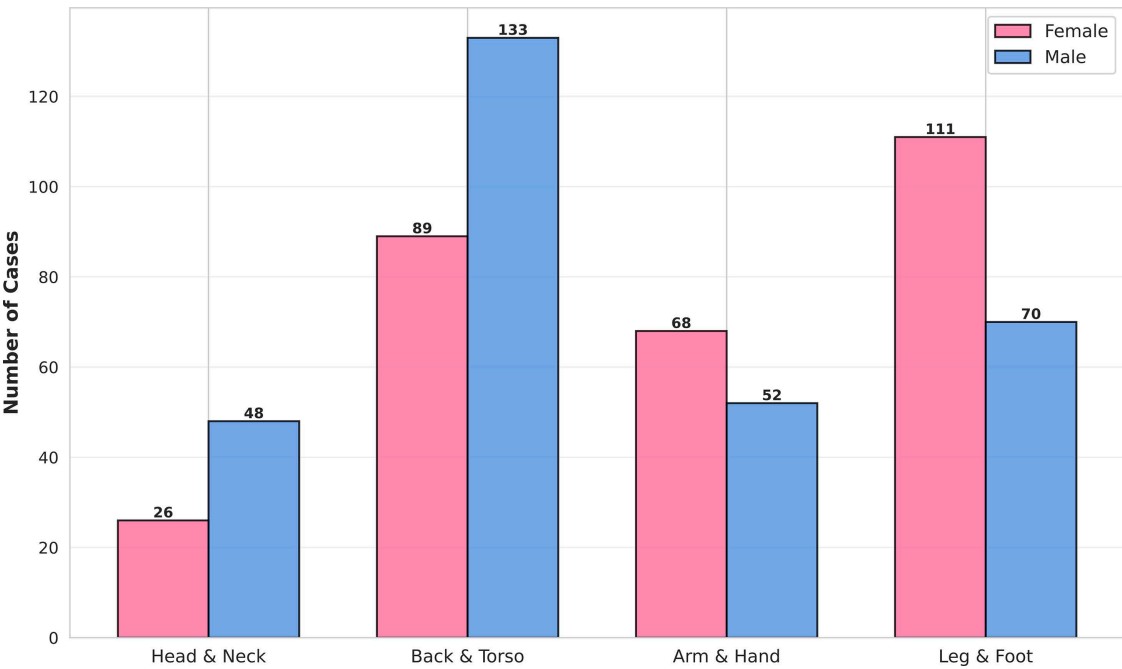

**Fig 4. Anatomical distribution of melanoma cases by sex.**

## 3.4 Subgroup analysis – MENA population

Among the 65 MENA patients identified in the cohort, melanoma presentation demonstrated distinct clinicopathological characteristics. Lebanese nationals accounted for the largest proportion of cases (40%), followed by Egyptians (16.9%), Emiratis (10.8%), Jordanians (7.7%), and Syrians (7.7%). The remaining nationalities (Palestinians, Bahrainis, Moroccans, Turks, and Omanis) contributed fewer than 5 cases. Sex distribution within the MENA subgroup showed a similar predominance (females 49.2%; males 50.8%), consistent with the overall sex distribution.

Histologic subtype analysis in the MENA subgroup demonstrated a predominance of SSM (58.5%), consistent with global patterns. The proportion of in situ melanomas in this subgroup was 16.9%, slightly higher than the overall cohort's (14.7%) (Fig 5).

Breslow thickness distribution in this subgroup of patients showed that 40% had thin melanomas (≤1.0 mm), indicating a favorable prognosis. An additional 30.8% had intermediate thickness (1.01–2.0 mm), whereas 20% and 9.2% had thicker tumors (2.01–4.0 mm and >4.0 mm). The mean thickness in this subgroup was 1.11 ± 2.42 mm, which was comparable to the overall cohort mean of 0.62 ± 1.03 mm (Mean Difference 0.49 mm; 95% CI [–0.10, 1.09]; $p = 0.10$)

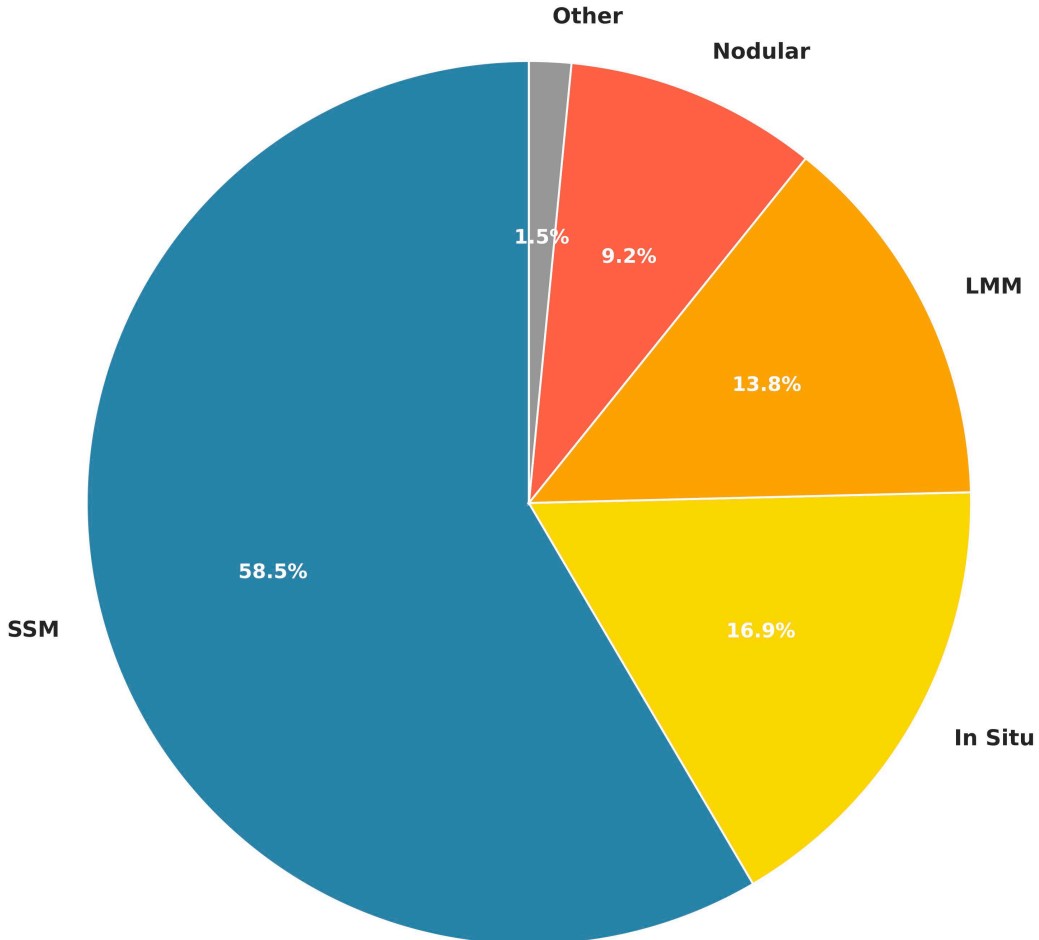

**Fig 5. Histologic subtype distribution of melanoma in the middle East and North African (MENA) subgroup.**

## 3.5 Predictors of thick melanoma

To identify independent factors associated with a higher risk of presenting with a thick melanoma (Breslow depth >1.0 mm), a multivariable logistic regression analysis was performed. The results, presented in **Table 3**, identified nodular histology as the strongest predictor, increasing the odds of a thick melanoma by over 18-fold compared to LMM (OR 18.40; 95% CI [7.08, 47.86]; $p < 0.001$). Other variables, including sex, age, and anatomical site, were not statistically significant independent predictors in this model.

## 4. Discussion

This eight-year, multi-ethnic study provides a comprehensive analysis of cutaneous melanoma in the UAE, revealing a significant burden of disease, particularly among the nation's large expatriate population. Our findings, which show a predominance of cases among individuals of European descent and distinct anatomical distribution patterns between sexes, reflect established global trends in the literature [10–12]. However, the primary significance of these data lies in spreading awareness among patients as well as physicians to screen for skin cancer, particularly melanoma.

### The burden of melanoma and global context

Melanoma has historically been considered rare in most African and Asian countries, including the Middle East, with incidence rates often below 1 per 100,000 person-years [2]. Our study's findings that 73.4% of melanoma cases occurred in individuals of European descent reflect the nation's unique demographic composition and the heightened vulnerability of fair-skinned expatriates to melanoma in the high-UV environment of the Arabian Peninsula.

The patterns observed in our study are consistent with established global trends regarding Breslow thickness, histological subtypes, and anatomical distribution. SSM was the most prevalent subtype, followed by melanoma in situ, LMM, and NM [1,11]. The distribution of Breslow thickness, with nearly 50% classified as thin (≤1.0 mm), is favorable, suggesting

**Table 3. Multivariable logistic regression of predictors of thick melanoma (Breslow thickness >1.0 mm). Nodular melanoma subtype and Clark level were the only significant independent predictors.**

| Variable | OR | 95% CI | P value |
|---|---|---|---|
| **Melanoma Type (Ref: LMM)** | | | |
| NM | 18.402 | 7.705–47.859 | <0.001 |
| SSM | 1.77 | 0.264–11.931 | 0.557 |
| Others | 2.145 | 0.518–8.879 | 0.295 |
| **Sex (Ref: females)** | | | |
| Male | 1.465 | 0.582–3.689 | 0.418 |
| **Age group (Ref: <30)** | | | |
| 30-39 | 0.259 | 0.043–1.558 | 0.14 |
| 40-49 | 0.958 | 0.177–5.192 | 0.961 |
| 50-59 | 0.447 | 0.076–2.642 | 0.374 |
| ≥60 | 0.958 | 0.165–5.559 | 0.962 |
| **Anatomical site (Ref: Leg & Foot)** | | | |
| Head and Neck | 1.537 | 0.357–6.623 | 0.564 |
| Back and Torso | 0.706 | 0.240–2.075 | 0.526 |
| Arm and Hand | 0.624 | 0.198–1.967 | 0.421 |

**Abbreviations:** Ref: Reference, OR: Odds Ratio, CI: Confidence Interval, LMM: Lentigo Maligna Melanoma, NM: Nodular Melanoma, SSM: Superficial Spreading Melanoma

some degree of early detection. However, this proportion remains below the 53.5% reported in countries with established public awareness programs [13]. The mean Breslow thickness of 0.62 mm in the overall cohort is comparable to that in developed nations; however, the significant difference between males and females warrants particular attention, as it reflects disparities in screening practices and disease detection patterns [10,14,15].

### Regional context: A comparison with the MENA subgroup

Our study provides a unique opportunity to contextualize the presentation of melanoma within the MENA region. A comparative analysis of our findings with data from Jordan and Iran reveals both similarities and important distinctions in the clinicopathological features of melanoma among different MENA populations. In our cohort, the mean Breslow thickness in the MENA subgroup was 1.12 ± 2.42 mm, which is notably lower than the 3.87 ± 3.35 mm reported by Malakoutikhah *et al.*[16]. This suggests that patients in the UAE may present with thinner tumors than those in other regions, potentially reflecting greater access to specialized dermatological care and higher health literacy in the UAE's metropolitan centers. However, the mean Breslow thickness in our MENA subgroup remains notably higher than the overall cohort's average of 0.62 ± 1.03 mm, indicating a higher-risk group.

Furthermore, the histological subtype distribution varies across the region. While SSM was the most common subtype in our overall cohort, ALM is the most prevalent subtype in Iran, accounting for 36.5% of cases in one study [16]. In contrast, ALM constituted only 3.7% of cases in our overall cohort and was not the predominant subtype in our MENA subgroup. This difference may be due to Iran's more conservative society with less exposure to sunlight and their darker skin phenotype. These differences further highlight the heterogeneity of melanoma presentation across the MENA region and underscore the importance of tailoring public health messaging to each country's specific epidemiological patterns.

### Sex and gender disparities in melanoma presentation

The higher incidence of melanoma in males and their presentation with thicker tumors is a well-documented phenomenon globally [10,14,15]. In our cohort, males presented at a mean age of 50.6 years, approximately 6.5 years older than females, and demonstrated a higher proportion of NM, a subtype associated with poor prognosis. The anatomical distribution also reflected sex-specific patterns: males showed a higher proportion of melanomas on the back and torso, and on the head and neck, whereas females predominantly presented with leg and foot lesions. These differences have been attributed to behavioral and occupational factors, with males being more likely to engage in outdoor activities without adequate sun protection and to work in occupations with prolonged sun exposure [15,17]. Furthermore, gender differences in healthcare-seeking behavior and attitudes toward preventive care contribute to later diagnosis among males [15].

### Cultural, environmental, and healthcare system barriers

The unique multicultural composition of the UAE, with its mix of sun-seeking expatriates and a local population that may have a false sense of security due to darker skin phenotypes, further complicates public health messaging and necessitates culturally appropriate, targeted interventions. The traditional full-body coverage in Emirati culture (traditionally called the kandura for males and the abaya for females), while providing sun protection, may paradoxically reduce awareness of skin changes and the importance of regular skin self-examination [5]. Furthermore, a cultural preference for tanned skin achieved through unprotected sunbathing contributes to reduced awareness of UV-related risks, particularly in regions with intense sunlight [5]. Additionally, there is a widespread misconception that skin cancer primarily affects individuals with lighter skin tones, leading to an underestimation of risk among darker-skinned populations. However, ALM, located on the palms, soles, and nail beds, accounts for a larger proportion of melanomas in individuals with skin of color. It is often misdiagnosed or missed as trauma or Tinea, treated for years before a biopsy reveals an invasive ALM [18,19].

## Evidence for the effectiveness of public health interventions

Overcoming these barriers requires a strategic shift from passive information dissemination to active, evidence-based public health engagement. The success of such initiatives is well documented in the international literature [13,20]. In countries with established awareness programs, a significant increase in the diagnosis of thin melanomas and a corresponding reduction in mortality have been observed. A retrospective analysis by Armstrong *et al.*[13] of a 10-year public awareness campaign demonstrated a substantial increase in the proportion of thin melanomas diagnosed, from 35.4% in 2003 to 53.5% in 2012, indicating a clear shift toward earlier detection [13]. Similarly, Matsumoto *et al.*[20] documented that a primary care-based melanoma screening initiative was associated with increased detection of thin melanoma and melanoma in situ, raising awareness of the importance of systematic screening.

The evidence for the efficacy of educational interventions is compelling. Nadratowski *et al.*[21] conducted a randomized online study to assess the impact of different social media messages on melanoma awareness (melanoma warning signs and correctly identifying moles) and skin self-examination behavior. Participants viewed Facebook and Instagram-style posts targeting either knowledge about melanoma, self-efficacy for skin checks, both, or control content. The study revealed that the group exposed to knowledge messages scored significantly higher on mole detection than the group exposed to control messages. Self-efficacy messages increased confidence and intentions to perform skin checks compared with control/knowledge-only messages [21]. These findings suggest that targeted, evidence-based social media content can positively influence both melanoma knowledge and self-screening behavior, offering a promising, scalable approach to enhance public engagement in melanoma prevention and early detection efforts.

## Leveraging digital health and social media for public engagement

In the modern digital landscape, social media and mobile health technologies offer an unprecedented opportunity to deliver scalable and cost-effective educational interventions. The growing influence of these platforms is transforming health communication globally. The study by Nadratowski *et al.*[21] provides a template for developing effective digital interventions. The integration of artificial intelligence (AI) for dermoscopic analysis and skin lesion classification could serve as a cost-effective and time-saving educational tool, allowing individuals to assess suspicious lesions before seeking professional evaluation, thereby advancing the frontiers of melanoma screening [22,23].

School-based interventions represent another clinical avenue for prevention. Guy *et al.*[24] demonstrated that primary and middle school-based programs can significantly improve sun-protection knowledge and behaviors, with effects persisting into adulthood. These programs should include supportive policies allowing sunscreen use, environmental modifications to increase shade availability, and comprehensive education on skin cancer risk and prevention. Such interventions are critical in the UAE, where a large proportion of the population is young and where early establishment of protective behaviors could have lifelong benefits.

Although our study does not explore the risk factors for melanoma. The UAE should include skin cancer in its health awareness campaigns. Based on the general evidence presented, we recommend a multi-pronged public health approach for the UAE:

1. National Awareness Campaign: Establishing a comprehensive, nationally coordinated skin cancer awareness campaign utilizing television, radio, print, and digital media. The campaign should be culturally sensitive, multilingual, and evidence-based.

2. Digital Health Integration: Develop and deploy mobile health applications and social media content targeting melanoma awareness and self-examination. These should incorporate interactive elements, dermoscopic images, and AI-assisted lesion assessment tools.

3. Healthcare System Integration: Integrate skin cancer screening into primary care settings and occupational health programs. Provide training to primary care physicians and occupational health nurses on melanoma recognition and referral protocols.

4. Targeted Interventions for High-Risk Groups: Develop targeted interventions for older adults, males, smokers, and uninsured populations to address the specific barriers these groups face in accessing screening.

5. School-Based Prevention: Implement comprehensive skin cancer prevention programs in primary and secondary schools, with an emphasis on sun protection and early detection.

6. Equity and Access: Ensure equitable access to dermatological expertise and screening services across all socio-economic strata and immigrant populations, addressing the social determinants of health that influence melanoma outcomes.

7. Data Collection and Registry: Establish a national melanoma registry to track incidence, outcomes, and the impact of public health interventions, enabling evidence-based refinement strategies over time.

## Limitations

This study poses several limitations. First, the dataset was derived from a single private tertiary referral center, which may introduce referral and selection bias and limit generalizability to the broader UAE population, particularly ndividuals with lower socioeconomic status who may have limited access to private specialized care. The cohort may therefore overrepresent individuals with greater healthcare access or fair-skinned expatriate populations. Additionally, the absence of a population denominator precludes estimation of melanoma incidence or prevalence, therefore, our findings describe clinicopathologic patterns among diagnosed cases rather than population-level epidemiology.

Second, the retrospective single-center tertiary design limited the availability of several established melanoma risk factors, including Fitzpatrick skin type, family history, UV exposure, and sun-protective behaviors. The absence of these variables limits the interpretation of demographic patterns and the ability to contextualize public health recommendations. Third, several clinically relevant oncologic variables were unavailable. Lymph node status and regional metastatic involvement were not captured because patients with invasive disease are typically referred to oncology centers for further staging and management. Molecular profiling, including BRAF mutation status, was also not routinely available. In addition, socioeconomic indicators were not captured, limiting the evaluation of potential disparities in healthcare access.

## 5. Conclusions

This study provides insight into the clinicopathological patterns of melanoma within a multi-ethnic cohort in the United Arab Emirates, contributing data from a region where melanoma remains understudied. While the findings highlight potential gaps in early detection and awareness in populations exposed to high ultraviolet radiation, these observations should be interpreted cautiously given the absence of behavioral and screening-related variables in the dataset. Future work should focus on multicenter studies integrating both public and private healthcare systems, as well as prospective investigations incorporating behavioral risk factors, socioeconomic indicators, screening practices, and molecular tumor profiling to better characterize melanoma risk and outcomes in the region.

## Supporting information

**S1 File. Strengthening the reporting of observational studies in epidemiology checklist.**
(PDF)

## Author contributions

**Conceptualization:** Jonathan Mokhtar.

**Data curation:** Jonathan Mokhtar, Rose Mary Eapen, Jeyaseelan Lakshmanan, Reem El-Bahtimi.

**Formal analysis:** Jeyaseelan Lakshmanan, Tom Loney.

**Investigation:** Rose Mary Eapen, Reem El-Bahtimi.

**Methodology:** Jonathan Mokhtar, Nada Alsuwaidi, Nada Hassane, Rose Mary Eapen, Hind Aljanaahi, Dalia AlDhamin, Tannaz Rahbari.

**Project administration:** Tom Loney, Reem El-Bahtimi.

**Resources:** Reem El-Bahtimi.

**Software:** Jonathan Mokhtar, Jeyaseelan Lakshmanan, Tom Loney.

**Supervision:** Tom Loney, Reem El-Bahtimi.

**Validation:** Reem El-Bahtimi.

**Visualization:** Jonathan Mokhtar, Ghazal Talal Saeed, Zainab Abdulla Al Darwish, Sara Almalik.

**Writing – original draft:** Jonathan Mokhtar, Nada Alsuwaidi, Nada Hassane, Hind Aljanaahi, Dalia AlDhamin, Tannaz Rahbari, Ghazal Talal Saeed, Zainab Abdulla Al Darwish, Sara Almalik.

**Writing – review & editing:** Jonathan Mokhtar, Nada Alsuwaidi, Nada Hassane, Rose Mary Eapen, Hind Aljanaahi, Dalia AlDhamin, Tannaz Rahbari, Ghazal Talal Saeed, Zainab Abdulla Al Darwish, Sara Almalik, Tom Loney.

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
