## [Decision Letter · Decision Letter 0]

12 Feb 2026

PONE-D-25-68202Epidemiological Patterns of Melanoma in a Multi-ethnic Cohort in the United Arab Emirates: 2017-2025PLOS One

Dear Dr. Mokhtar,

Thank you for submitting your manuscript to PLOS ONE. After careful consideration, we feel that it has merit but does not fully meet PLOS ONE’s publication criteria as it currently stands. Therefore, we invite you to submit a revised version of the manuscript that addresses the points raised during the review process.

We look forward to receiving your revised manuscript.

Kind regards,

Mohamed Gouda, MD, PhD

Academic Editor

PLOS One

Journal Requirements:

2.  In the online submission form, you indicated that “The data underlying the results presented in the study are available from the corresponding author upon reasonable request and approval from the Dubai Research Scientific Research Ethics Committee approval.”

**Additional Editor Comments:**

REVIEWER 1

This manuscript provides valuable clinicopathological insights into cutaneous melanoma in a multi-ethnic cohort from the United Arab Emirates, a region with limited published data. The relatively large sample size, centralized histopathological confirmation, and analysis of sex- and subtype-specific tumor characteristics are notable strengths. Several issues should be addressed to improve scientific clarity and interpretability.

First, the study design does not allow estimation of melanoma prevalence or incidence, as no population denominator is available. References to epidemiological burden or population-level measures should therefore be revised, with findings consistently framed as distributions and patterns among diagnosed cases.

Second, the single-center, tertiary referral nature of the dataset introduces potential selection and referral bias, likely overrepresenting fair-skinned expatriates and individuals with greater healthcare access. This limitation should be more explicitly acknowledged, and conclusions regarding national representativeness should be appropriately tempered.

Third, the multivariable logistic regression model includes Clark level as an independent predictor of Breslow thickness, although both variables reflect tumor depth and are biologically correlated. The very high odds ratios observed may therefore reflect intrinsic histopathological correlation rather than independent predictive value, and this should be justified or interpreted with caution.

Importantly, several established melanoma risk factors were not captured, including skin phenotype or skin color, family history of melanoma, occupational or recreational ultraviolet exposure, and sunscreen use. The absence of these variables limits interpretation of demographic patterns and restricts the ability to contextualize public health recommendations.

Finally, while the discussion emphasizes the need for awareness and screening initiatives, the lack of data on sun-protective behaviors and screening practices necessitates framing these conclusions as contextual and hypothesis-generating rather than directly supported by the study findings. Addressing these points will substantially strengthen the rigor, transparency, and impact of the manuscript.

REVIEWER 2

This a single-center retrospective analysis evaluating cutaneous melanoma cases at a tertiary referral centre between 2017-2025, including 597 patients with in situ and invasive melanoma. The study is very well written and addresses important data re: the diagnosis of melanoma in the UAE. Results are largely influenced by the disproportionate predominance of European ethnicity. However, I have some observations regarding this study that should be addressed before the study is considered for publication:

1. Authors report pathological data including Breslow thickness, and Clark Level. However, authors did not report mitotic rate, presence of ulceration, vascular or perineural invasion. This should be considered to be included as considerably limits the validity of this analysis.

2. Authors did not report lymph node status as such this is as important limitation and missed opportunity that would strenghten this manuscript considerably. If metastatic tumors were excluded, I strongly recommend authors to include lymph node status and to report regional and locoregional involvement as per AJCC staging.

3. More information re: pathological assessment should be provided. For instance, is molecular assessment performed in this jurisdiction? BRAF status alongside common mutations in cutaneous melanoma should be described.

4. Socioeconomic status of population diagnosed if available should be included as its limits interpretation. For instance, is postal code available? If so this would also improve readers interprepation of study results.

5. Authors include Clark Level as an independent variable to predict melanoma thickness. Clark levels are correlated therefore it doesn't make much sense to include both, and thus Clark level should be removed from regression model.

6. Authors discussion is highly based in public health strategies to raise awareness of melanoma. This is interesting, however given the remarkably limited granularity of the data reported in its present form if not revised should be tempered as well as significantly trimmed as it goes beyond this original research MS. Likewise, largely the abovementioned concerns are not acknowledged in the limitations, and thus should be revised.

REVIEWER 3

The manuscript is a valuable and timely contribution that addresses a critical gap in the epidemiology of cutaneous melanoma in the UAE and the broader MENA region. The large, multi-ethnic cohort, adherence to STROBE guidelines, centralized histopathological review, and comprehensive statistical analyses are notable strengths. I recommend acceptance pending minor revisions to further strengthen the scientific rigor, clarity, and impact of the work. First, the Introduction and Discussion would benefit from substantial condensation to reduce redundancy and improve narrative focus, particularly where global data and public health strategies are reiterated. Second, clearer justification and methodological detail are required for key analytical decisions, including the treatment of Clark level as a continuous variable and the selection of covariates in the multivariable model, to enhance reproducibility and transparency. Third, the limitations section should be expanded to more explicitly address referral bias, representativeness of a single private tertiary center, and potential misclassification related to ethnicity and access to care. Addressing these points will substantially enhance the manuscript’s clarity, methodological robustness, and suitability for publication.

Reviewers' comments:

Reviewer's Responses to Questions

**Comments to the Author**

1. Is the manuscript technically sound, and do the data support the conclusions?

Reviewer #1: Yes

Reviewer #2: Partly

Reviewer #3: Yes

2. Has the statistical analysis been performed appropriately and rigorously? 

Reviewer #1: I Don't Know

Reviewer #2: N/A

Reviewer #3: Yes

3. Have the authors made all data underlying the findings in their manuscript fully available?

Reviewer #1: No

Reviewer #2: No

Reviewer #3: Yes

4. Is the manuscript presented in an intelligible fashion and written in standard English?

Reviewer #1: Yes

Reviewer #2: Yes

Reviewer #3: Yes

5. Review Comments to the Author

Reviewer #1: This manuscript provides valuable clinicopathological insights into cutaneous melanoma in a multi-ethnic cohort from the United Arab Emirates, a region with limited published data. The relatively large sample size, centralized histopathological confirmation, and analysis of sex- and subtype-specific tumor characteristics are notable strengths. Several issues should be addressed to improve scientific clarity and interpretability.

First, the study design does not allow estimation of melanoma prevalence or incidence, as no population denominator is available. References to epidemiological burden or population-level measures should therefore be revised, with findings consistently framed as distributions and patterns among diagnosed cases.

Second, the single-center, tertiary referral nature of the dataset introduces potential selection and referral bias, likely overrepresenting fair-skinned expatriates and individuals with greater healthcare access. This limitation should be more explicitly acknowledged, and conclusions regarding national representativeness should be appropriately tempered.

Third, the multivariable logistic regression model includes Clark level as an independent predictor of Breslow thickness, although both variables reflect tumor depth and are biologically correlated. The very high odds ratios observed may therefore reflect intrinsic histopathological correlation rather than independent predictive value, and this should be justified or interpreted with caution.

Importantly, several established melanoma risk factors were not captured, including skin phenotype or skin color, family history of melanoma, occupational or recreational ultraviolet exposure, and sunscreen use. The absence of these variables limits interpretation of demographic patterns and restricts the ability to contextualize public health recommendations.

Finally, while the discussion emphasizes the need for awareness and screening initiatives, the lack of data on sun-protective behaviors and screening practices necessitates framing these conclusions as contextual and hypothesis-generating rather than directly supported by the study findings. Addressing these points will substantially strengthen the rigor, transparency, and impact of the manuscript.

Reviewer #2: This a single-center retrospective analysis evaluating cutaneous melanoma cases at a tertiary referral centre between 2017-2025, including 597 patients with in situ and invasive melanoma. The study is very well written and addresses important data re: the diagnosis of melanoma in the UAE. Results are largely influenced by the disproportionate predominance of European ethnicity. However, I have some observations regarding this study that should be addressed before the study is considered for publication:

1. Authors report pathological data including Breslow thickness, and Clark Level. However, authors did not report mitotic rate, presence of ulceration, vascular or perineural invasion. This should be considered to be included as considerably limits the validity of this analysis.

2. Authors did not report lymph node status as such this is as important limitation and missed opportunity that would strenghten this manuscript considerably. If metastatic tumors were excluded, I strongly recommend authors to include lymph node status and to report regional and locoregional involvement as per AJCC staging.

3. More information re: pathological assessment should be provided. For instance, is molecular assessment performed in this jurisdiction? BRAF status alongside common mutations in cutaneous melanoma should be described.

4. Socioeconomic status of population diagnosed if available should be included as its limits interpretation. For instance, is postal code available? If so this would also improve readers interprepation of study results.

5. Authors include Clark Level as an independent variable to predict melanoma thickness. Clark levels are correlated therefore it doesn't make much sense to include both, and thus Clark level should be removed from regression model.

6. Authors discussion is highly based in public health strategies to raise awareness of melanoma. This is interesting, however given the remarkably limited granularity of the data reported in its present form if not revised should be tempered as well as significantly trimmed as it goes beyond this original research MS. Likewise, largely the abovementioned concerns are not acknowledged in the limitations, and thus should be revised.

Reviewer #3: The manuscript is a valuable and timely contribution that addresses a critical gap in the epidemiology of cutaneous melanoma in the UAE and the broader MENA region. The large, multi-ethnic cohort, adherence to STROBE guidelines, centralized histopathological review, and comprehensive statistical analyses are notable strengths. I recommend acceptance pending minor revisions to further strengthen the scientific rigor, clarity, and impact of the work. First, the Introduction and Discussion would benefit from substantial condensation to reduce redundancy and improve narrative focus, particularly where global data and public health strategies are reiterated. Second, clearer justification and methodological detail are required for key analytical decisions, including the treatment of Clark level as a continuous variable and the selection of covariates in the multivariable model, to enhance reproducibility and transparency. Third, the limitations section should be expanded to more explicitly address referral bias, representativeness of a single private tertiary center, and potential misclassification related to ethnicity and access to care. Addressing these points will substantially enhance the manuscript’s clarity, methodological robustness, and suitability for publication.

6. PLOS authors have the option to publish the peer review history of their article (what does this mean?). If published, this will include your full peer review and any attached files.

Reviewer #1: **Yes:**Omnia Korani

Reviewer #2: No

Reviewer #3: **Yes:**Firas Kreidieh

---

## [Author Response · Author response to Decision Letter 1]

25 Mar 2026

PLOS One Response to Reviewer Form

March 13th, 2026

Re: Clinicopatholoical Features and Ethnic Disparities of Melanoma in the United Arab Emirates: 2017-2025

PONE-D-25-68202

Dear Reviewers,

Thank you for taking the time to review our manuscript. We appreciate your valuable feedback and the opportunity to improve our work.

We have carefully considered all the comments and suggestions provided. Below, we outline our responses to each point raised by the reviewers in the letter dated February 12th, 2026, along with the corresponding revisions to the manuscript in marked-up and clean versions.

We sincerely appreciate your insightful input, which has helped strengthen our study. Please find our detailed responses below.

Reviewer Comment Author Comments Manuscript Revisions

Reviewer 1:

Comment 1

This manuscript provides valuable clinicopathological insights into cutaneous melanoma in a multi-ethnic cohort from the United Arab Emirates, a region with limited published data. The relatively large sample size, centralized histopathological confirmation, and analysis of sex- and subtype-specific tumor characteristics are notable strengths. Several issues should be addressed to improve scientific clarity and interpretability.

We thank the reviewer for their positive feedback. No revisions required.

Comment 2

First, the study design does not allow estimation of melanoma prevalence or incidence, as no population denominator is available. References to epidemiological burden or population-level measures should therefore be revised, with findings consistently framed as distributions and patterns among diagnosed cases. We appreciate this important methodological clarification. As correctly noted by the reviewer, our dataset is derived from diagnosed cases at a referral dermatopathology center and therefore lacks a population denominator necessary to estimate incidence or prevalence.

To address this, we revised the manuscript to ensure that our findings are framed strictly as clinicopathological distributions among diagnosed melanoma cases rather than population-level epidemiological measures. Revisions made in the title- page 1, and throughout the manuscript to remove the word or implications of “epidemiology”.

Comment 3

Second, the single-center, tertiary referral nature of the dataset introduces potential selection and referral bias, likely overrepresenting fair-skinned expatriates and individuals with greater healthcare access. This limitation should be more explicitly acknowledged, and conclusions regarding national representativeness should be appropriately tempered. We agree with the reviewer that referral bias is an important limitation in our study. Because the dataset originates from a specialized dermatopathology referral laboratory, the cohort may overrepresent individuals with greater healthcare access or those referred for specialized evaluation.

To address this, we expanded the limitations section and tempered the interpretation of our findings throughout the discussion. Revisions in discussion section pages 13-16 and limitation section page 17 lines 552-568.

Comment 4

Third, the multivariable logistic regression model includes Clark level as an independent predictor of Breslow thickness, although both variables reflect tumor depth and are biologically correlated. The very high odds ratios observed may therefore reflect intrinsic histopathological correlation rather than independent predictive value, and this should be justified or interpreted with caution. We thank the reviewer for this important methodological observation. Clark level and Breslow thickness are indeed biologically correlated measures of tumor invasion depth.

In response, we removed Clark level from the multivariable logistic regression model. Revisions made by removing clark level from multivariable logistic regression models from methodology section page 6, 2.6 statistical analysis lines 227-235, results section page 11, 3.4 (predictors of thick melanoma) and table 3 (page 12).

Comment 5

Importantly, several established melanoma risk factors were not captured, including skin phenotype or skin color, family history of melanoma, occupational or recreational ultraviolet exposure, and sunscreen use. The absence of these variables limits interpretation of demographic patterns and restricts the ability to contextualize public health recommendations. We agree that these variables are critical for understanding melanoma risk patterns. However, these data were not available in the dermatopathology lab records used in this study, as the dataset did not include detailed clinical histories or patient interviews.

To address this limitation, we acknowledged the absence of these variables in the limitations section.

Revisions made in limitations section page 17 – lines 560-564.

Comment 6

Finally, while the discussion emphasizes the need for awareness and screening initiatives, the lack of data on sun-protective behaviors and screening practices necessitates framing these conclusions as contextual and hypothesis-generating rather than directly supported by the study findings. Addressing these points will substantially strengthen the rigor, transparency, and impact of the manuscript. We appreciate this comment and agree that recommendations regarding melanoma awareness campaigns should be interpreted cautiously given the absence of behavioral or screening data.

Accordingly, we revised the conclusion to ensure that these interpretations are clearly framed as contextual observations and hypothesis-generating rather than direct conclusions supported by the datasets. Revision in conclusion section page 18.

Reviewer 2:

Comment 1

This a single-center retrospective analysis evaluating cutaneous melanoma cases at a tertiary referral centre between 2017-2025, including 597 patients with in situ and invasive melanoma. The study is very well written and addresses important data re: the diagnosis of melanoma in the UAE. Results are largely influenced by the disproportionate predominance of European ethnicity. However, I have some observations regarding this study that should be addressed before the study is considered for publication: We thank the reviewer for their positive and supportive feedback. No revisions required.

Comment 2

Authors report pathological data including Breslow thickness, and Clark Level. However, authors did not report mitotic rate, presence of ulceration, vascular or perineural invasion. This should be considered to be included as considerably limits the validity of this analysis. We thank the reviewer for highlighting this important detail. We revisited the original pathology reports and extracted additional histopathological features including:

1. Ulceration

2. Mitotic activity

3. Lymphovascular invasion (LVI)

4. Perineural invasion (PNI). Revisions made in methodology section 2.4 (data collection and variable definitions) page 5, lines 73-78, results section (outcome data), page 9 lines 262-266 and Table 2 (pages 9-10).

Comment 3

Authors did not report lymph node status as such this is as important limitation and missed opportunity that would strenghten this manuscript considerably. If metastatic tumors were excluded, I strongly recommend authors to include lymph node status and to report regional and locoregional involvement as per AJCC staging. We agree that lymph node status is an important prognostic variable in melanoma staging. However, our dataset is derived from a dermatopathology referral laboratory and therefore contains only the primary diagnostic pathology reports.

Patients diagnosed with invasive melanoma are typically referred to specialized oncology centers for: sentinel lymph node biopsy, staging, surgical/oncologic management.

Nonetheless, we expanded the limitations section to include this. Revisions made in limitations section, page 17, lines 563-568.

Comment 4

More information re: pathological assessment should be provided. For instance, is molecular assessment performed in this jurisdiction? BRAF status alongside common mutations in cutaneous melanoma should be described. We thank the reviewer for this suggestion. Molecular testing for melanoma is typically performed outside the UAE and is ordered by oncologists after referral, hence why we could not have this data in our study.

We have expanded the limitation section to include this. Revisions made in limitations section, page 17, lines 563-568.

Comment 5

Socioeconomic status of population diagnosed if available should be included as its limits interpretation. For instance, is postal code available? If so this would also improve readers interprepation of study results. We agree that socioeconomic indicators could provide important context for understanding healthcare access and screening behaviors. However, these data were not available in the pathology database used in this study, and the dataset did not include postal codes (postal codes aren’t used in the UAE) or other socioeconomic indicators.

This has been mentioned in the limitations section. Revisions made in limitations section, page 17, lines 567-568.

Comment 6

Authors include Clark Level as an independent variable to predict melanoma thickness. Clark levels are correlated therefore it doesn't make much sense to include both, and thus Clark level should be removed from regression model. We thank the reviewer for this important observation. We do agree that clark levels are correlated and are a well-known factor, hence we have removed it from the multivariable logistic regression models. Revisions made by removing clark level from multivariable logistic regression models from methodology section page 6, 2.6 statistical analysis lines 227-235, results section page 11, 3.4 (predictors of thick melanoma) and table 3 (page 12).

Comment 7

Authors discussion is highly based in public health strategies to raise awareness of melanoma. This is interesting, however given the remarkably limited granularity of the data reported in its present form if not revised should be tempered as well as significantly trimmed as it goes beyond this original research MS. Likewise, largely the abovementioned concerns are not acknowledged in the limitations, and thus should be revised.

We appreciate this suggestion and have revised the manuscript accordingly, by trimming down the discussion to avoid redundancy, focusing on interpretation of the key findings.

We also had focused on targeted public health strategies and awareness campaigns as the UAE lacks this fundamental aspect of skin cancer screening and with the utilization of modern day technology we aimed to provide a base for future trials and studies incorporating these techniques for increased awareness. Revisions made throughout the discussion and limitations section – pages 13-17, lines 381-568.

Reviewer 3:

Comment 1

The manuscript is a valuable and timely contribution that addresses a critical gap in the epidemiology of cutaneous melanoma in the UAE and the broader MENA region. The large, multi-ethnic cohort, adherence to STROBE guidelines, centralized histopathological review, and comprehensive statistical analyses are notable strengths. I recommend acceptance pending minor revisions to further strengthen the scientific rigor, clarity, and impact of the work. We thank the reviewer for their supportive feedback and look forward to this paper being published in PONE. No revisions required.

Comment 2

First, the Introduction and Discussion would benefit from substantial condensation to reduce redundancy and improve narrative focus, particularly where global data and public health strategies are reiterated.

We appreciate this suggestion and have revised the manuscript accordingly, by trimming down the discussion to avoid redundancy (leaving the parts discussed in the introduction only in the introduction section and removing it from the discussion), focusing on interpretation of the key findings.

We also had focused on targeted public health strategies and awareness campaigns as the UAE lacks this fundamental aspect of skin cancer screening and with the utilization of modern day technology we aimed to provide a base for future trials and studies incorporating these techniques for increased awareness. Revisions made throughout the discussion and limitations section – pages 13-17, lines 381-568.

Comment 3

Second, clearer justification and methodological detail are required for key analytical decisions, including the treatment of Clark level as a continuous variable and the selection of covariates in the multivariable model, to enhance reproducibility and transparency. Following the reviewers’ concerns and other reviewers, Clark level was removed from the regression model entirely to avoid collinearity with Breslow thickness.

Revisions made by removing clark level from multivariable logistic regression models from methodology section page 6, 2.6 statistical analysis lines 227-235, results section page 11, 3.4 (predictors of thick melanoma) and table 3 (page 12).

Comment 4

Third, the limitations section should be expanded to more explicitly address referral bias, representativeness of a single private tertiary center, and potential misclassification related to ethnicity and access to care. Addressing these points will substantially enhance the manuscript’s clarity, methodological robustness, and suitability for publication. The limitations section was substantially expanded to address the following:

1. Referral bias

2. Representativeness

3. Missing behavioral variables

4. Absence of staging data

5. Lack of socioeconomic data

6. Lack of molecular testing

These additions improve transparency and interpretability. Revisions made in limitations section page 17, lines 552-568.

Thank you once again for your time and consideration. We look forward to your feedback.

Kind regards,

Jonathan Mokhtar

MD Candidate (Class of 2026)

College of Medicine, Mohammed Bin Rashid University of Medicine and Health Sciences

Dubai, United Arab Emirates

---

## [Decision Letter · Decision Letter 1]

17 May 2026

Clinicopathological Features and Ethnic Disparities of Melanoma in the United Arab Emirates: 2017-2025

PONE-D-25-68202R1

Dear Dr. Mokhtar,

We’re pleased to inform you that your manuscript has been judged scientifically suitable for publication and will be formally accepted for publication once it meets all outstanding technical requirements.

Kind regards,

Mohamed Gouda, MD, PhD

Academic Editor

PLOS One

Additional Editor Comments (optional):

Reviewers' comments:

Reviewer's Responses to Questions

**Comments to the Author**

1. If the authors have adequately addressed your comments raised in a previous round of review and you feel that this manuscript is now acceptable for publication, you may indicate that here to bypass the “Comments to the Author” section, enter your conflict of interest statement in the “Confidential to Editor” section, and submit your "Accept" recommendation.

Reviewer #1: All comments have been addressed

2. Is the manuscript technically sound, and do the data support the conclusions?

Reviewer #1: Yes

3. Has the statistical analysis been performed appropriately and rigorously? 

Reviewer #1: Yes

4. Have the authors made all data underlying the findings in their manuscript fully available?

Reviewer #1: Yes

5. Is the manuscript presented in an intelligible fashion and written in standard English?

Reviewer #1: Yes

6. Review Comments to the Author

Reviewer #1: I would like to thank the authors for their efforts in revising the manuscript and for adequately addressing the reviewers’ comments. The revisions have improved the clarity and overall quality of the work. I have no further major concerns.

7. PLOS authors have the option to publish the peer review history of their article (what does this mean?). If published, this will include your full peer review and any attached files.

Reviewer #1: No

---

## [Editor Report · Acceptance letter]

PONE-D-25-68202R1

PLOS One

Dear Dr. Mokhtar,

I'm pleased to inform you that your manuscript has been deemed suitable for publication in PLOS One. Congratulations! Your manuscript is now being handed over to our production team.

Kind regards,

on behalf of

Dr. Mohamed Gouda

Academic Editor

PLOS One